# Laboratory Experiments to Assess the Effect of Chlorella on Turbidity Estimation

Wenxiang Zhang [1], Dan Zhang [1], Benwei Shi [1,2,*], Zhonghao Zhao [1], Jianxiong Sun [1], Yujue Wang [1], Xing Wang [3], Yang Lv [4], Yue Li [1] and Youcai Liu [5,*]

1   State Key Laboratory of Estuarine and Coastal Research, East China Normal University, Shanghai 200241, China
2   Key Laboratory of Coastal Science and Integrated Management, Ministry of Natural Resources, Qingdao 266061, China
3   Chinese Research Academy of Environmental Sciences, Beijing 100012, China
4   Beijing Aisiyi Technology Co., Ltd., Beijing 100102, China
5   Hebei Hydrological Engineering Geological Exploration Institute, Yuhua District, Shijiazhuang 050051, China
*   Correspondence: bwshi@sklec.ecnu.edu.cn (B.S.); skyhks@163.com (Y.L.)

**Abstract:** Turbidity is an important parameter in monitoring water quality, and thus attracts significant attention. Changes in the various components and constituent elements of water directly affect turbidity measurements. The turbidity of water is generally quantified by measuring the absorbance or scattering characteristics of substances suspended in it. The complex composition of environmental water bodies complicates the determination of factors influencing their turbidity. Controlled experiments that quantitatively analyze the effect of a single factor on the measurement of turbidity in the laboratory are an important means to improve the accuracy of turbidity assessment. Properties of suspended materials in a water column that may affect its measured turbidity include the concentration of algae, particle size, and the color of soluble substances, etc. The laboratory experiments reported here used Chlorella as an example to investigate the effect of algal concentration on turbidity measurement. The results are as follows. When the turbidity is low (100 NTU), the average relative error between the theoretical and practical absorbance is about 37.52%, which decreases to 19.20% at 100–200 NTU and 5.16% at 200–400 NTU. The characteristic spectral bands sensitive to turbidity (680 nm) and Chlorella (240 nm) were selected, and the theoretical and practical turbidity results were analyzed. The average relative errors of mixed liquids of less than 100, 100–200, and 200–400 NTU are 65.07%, 34.18%, and 3.95%, respectively. Therefore, the concentration of Chlorella significantly affects the measured turbidity, and results in a more complex effect at low turbidity (<100 NTU). Combining the analysis of absorbance peak values and characteristic spectral bands, we can assess the turbidity changes in different components, and through calibration, information regarding the concentration and variation of different components in water bodies can be obtained. The results of this research could improve the accuracy of on-site measurement of the concentrations of different components suspended in water, and also facilitate the development of new turbidity sensors.

**Keywords:** turbidity; chlorella; water quality; modeling; spectral frequency

## 1. Introduction

Turbidity is an important parameter in water quality measurement and a comprehensive indicator of water quality [1–3]. It can indicate the effects of runoff from construction, agriculture, logging, and other sources of discharge. Environmental monitoring and factor analysis of rivers, lakes, estuaries, and coastal waters all need to measure turbidity [4,5]. Turbidity is a physical property of fluids related to reduced optical transparency due to the presence of suspended and dissolved materials [6]. These materials can be of organic or inorganic origin, vary in color, composition, and size, and are typically in the range of

0.004 mm (clay) to 1.0 mm (sand) [7]. Turbidity measurements are often used to indicate water quality based on clarity and estimated total suspended material. By calibrating the turbidity of a water column, information regarding the concentration and changes in the suspended materials can be obtained [6,8]. These suspended materials are closely related to the transport of nutrients and the diffusion of pollutants, making on-site turbidity estimation particularly useful for scientific research and environmental monitoring. For example, eutrophication and rapid plankton growth will increase the turbidity of water, which will depend on the particle size and concentrations of algae and other suspended materials [1,9]. Therefore, it would be helpful to identify the effects of particular suspended matter components on turbidity to use turbidity measurements for the assessment of those materials in environmental samples.

Measuring the absorbance of suspended matter is one of the primary methods used to assess turbidity. A significant amount of research has considered the methodologies of turbidity measurement and algae identification [10–13]. Monitoring and error analysis of chlorophyll concentrations in algae have assessed the effects of algae and other components on optical sensor measurement [14–19]. Uncertainty analysis and improvements in algorithms have facilitated estimation of the concentrations of different types of algae in water [20–24]. Previous studies have focused primarily on the effect of factors, such as turbidity, pH, and temperature on the measurement of different types of algae in water [25,26]. Some works have analyzed the scattering or reflection of a specific component to obtain the changes in turbidity associated with that component [27,28]. However, different constituents and compositions of algae, particles or colors in water might affect turbidity measurements in ways that are not easily discerned, making the factors affecting turbidity measurement in complex environmental water bodies difficult to control [29]. Controlled laboratory experiments represent an important means of quantitatively analyzing the influences of single factors on turbidity measurements, and improve the accuracy of measuring turbidity attributable to different components in water [12,30].

The turbidity of water is related to the absorbance and scattering of light by materials suspended in it. The absorbance and scattering, inherent optical parameters, are related to the optical characteristics of the components in the water column, and they can only be obtained through laboratory experiments and calculations [31]. Absorbance peaks of the same substances at different concentrations and the corresponding spectral changes can be used as the basis for quantitative analysis. Due to the complexity of water components, the absorption curves, maximum absorption peaks, and spectra of different components will be different. This work considered Chlorella and standard turbidity solutions as examples, and experimentally assessed water with Chlorella, various standard turbidity solutions, and mixtures of both to obtain their inherent optical parameters of absorbance peak values and spectral changes. In this article, a certain concentration of Chlorella (a genus of green algae) and various standard turbidity liquids were selected. The effect of Chlorella on turbidity was discussed by analyzing the absorbance of univariate and mixed liquids and evaluating measurement errors. This paper aimed to quantitatively analyze the impact of algae on turbidity measurement, improve the accuracy of turbidity measurement of water with suspended particle components, and provide technical support through calibration for the accurate assessment of suspended particle concentrations in environmental water samples.

## 2. Materials and Methods

### 2.1. Experimental Equipment and Materials

This experiment was carried out by spectrophotometry (UV-4800; spectral range, 190–1100 nm, Unico (Shanghai) Co., Ltd., Shanghai, China) and the spectrophotometer was tested and calibrated by the manufacturer. Standard turbidity solutions were used (0–4000 NTU, HACH Co., Ltd., Loveland, CO, USA). Chlorella, a representative genus of green algae (Figure 1), was used to analyze the effects of algae on turbidity measurement. Optical microscope (Olympus Co., Ltd., Tokyo, Japan) counting assessed the concentration of Chlorella and obtained the magnification of microscope imaging of 40 times (Figure 1).

Nine groups of experimental samples were prepared: One containing $10^7$ N/L of Chlorella (with N as the number of Chlorella), the quantity of $10^7$ Chlorella per liter and eight standard turbidity solutions (samples 1–8) (Table 1).

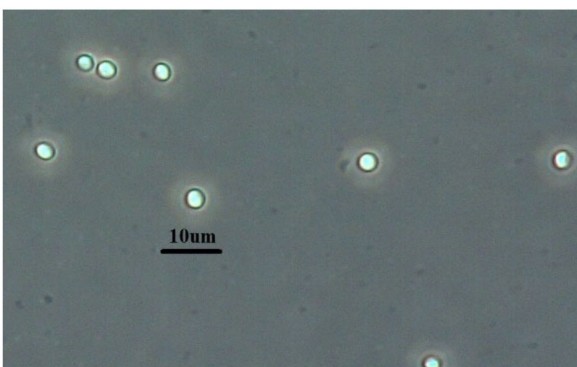

**Figure 1.** Particle size distribution of Chlorella (light microscope image at 40× magnification).

**Table 1.** Turbidity of standard solutions (NTU).

| Samples | 1 | 2 | 3 | 4 | 5 | 6 | 7 | 8 |
|---|---|---|---|---|---|---|---|---|
| Turbidity | 50 | 100 | 150 | 200 | 250 | 300 | 350 | 400 |

The standard turbidity solutions were prepared using the following standard of ISO 7027 [32]. First, we soaked a membrane filter of pore size 0.45 um for 1 h in 100 mL of distilled water. Filter 250 mL of distilled water through it and discarded the water. Thereafter, we passed a 2-L volume of distilled water two times through the membrane and reserved this water for the preparation of the formazine suspensions. Other particle-free waters, such as reverse osmosis water, can be used instead. Second, we diluted the standard turbidity solution in proportion according to the experimental needs. Finally, a 10 mL aliquot of the Chlorella sample with known concentration was mixed with a set amount (10 mL) of the eight standard turbidity solutions. The absorbances of the Chlorella sample, the eight standard solutions (Table 1), and the eight mixed solutions were measured. During the experiment, each sample was uniformly mixed, and the number and time were unified. To reduce the human error in the measurement process, three groups of stable data were selected from five groups of data for average and experimental analysis.

*2.2. Absorbance Theory*

Absorbance was used here to analyze the magnitude and change in turbidity [6,33], and is described by the Beer–Lambert law. A theoretical analysis model was established based on the principle of absorbance as an additive. The absorbance (*A*) of a liquid is inversely proportional to its transmittance (the ratio of the transmitted light intensity to the incident light intensity):

$$A = -lg(\frac{I_1}{I_0}) \tag{1}$$

where $I_0$ is the intensity of incident light, and $I_1$ is the intensity of transmitted light.

The Beer–Lambert law states

$$A = \varepsilon l C \tag{2}$$

where $\varepsilon$ is the molar absorption coefficient [L/(μmol·cm)], $l$ is the optical path (cm), and $C$ is the sample concentration (μmol/L). Equations (1) and (2) provide the following

$$C = \frac{1}{\varepsilon l} lg \frac{I_0}{I_1} \tag{3}$$

where $1/\varepsilon l$ is the slope of the calibration curve for a specific optical path. Once the slope is known, Equation (3) allows the concentration of a sample to be calculated from the measured light intensity passing through it (before and after) [34].

## 3. Results

### 3.1. The Variation of Turbidity

Changes in turbidity are closely related to changes in absorbance and spectral band properties. Adding Chlorella to the standard turbidity solutions induced changes in the resulting mixed liquids' absorbance peaks. With increasing turbidity, the corresponding absorbance peak values increased gradually (Figure 2). The standard turbidity solutions (Figure 2a) showed two absorbance peaks at low turbidity (≤100 NTU): The 50 NTU solution showed peaks of 1.323 and 1.003, respectively, and the 100 NTU solution showed peaks of 1.774 and 1.811, respectively. At greater turbidity (>100 NTU), the standard solutions showed only one absorbance peak. Adding Chlorella (Figure 2b) caused the double peaks of the original solutions to be replaced by single peaks. The absorbance peaks ranged between 1.51 and 2.70 at spectral bands of 200–246 nm. Moreover, the peak values of spectral bands of the mixed solution increased with the growing turbidity.

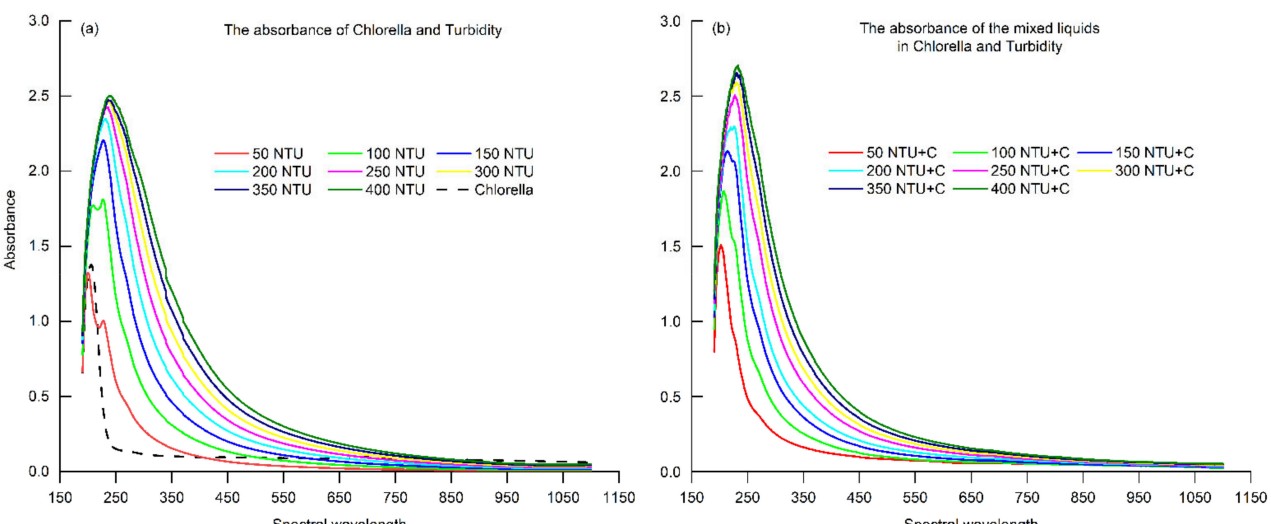

**Figure 2.** Absorbance of Chlorella and standard turbidity solutions (**a**) and mixtures of Chlorella and standard turbidity solutions (**b**).

### 3.2. Turbidity Calibration

To study the effect of Chlorella on turbidity measurement, a model of the relationship between turbidity and absorbance was established. Correlation analysis compared the changes in turbidity and the corresponding absorbance (Figure 3a–c). The absorbance correlation of the mixed liquids (Figure 3c) was higher than the standard turbidity liquids (Figure 3a,b). The absorbance of the characteristic 680 nm spectrum—which is sensitive to turbidity—was specifically considered by the model (Figure 4). Table 2 shows the error analysis. For the mixtures, the calculations used the theoretical and measured spectra at 680 nm. Table 2 shows that the total average relative error (Er-A-680 nm) between the theoretical and experimental absorbance at 680 nm is ~50%. Below 150 NTU, the relative error of absorbance exceeded 50% (the maximum was 53.41%); for 150–400 NTU, the relative error was 45–50% (Table 2). If the peak values of the absorbance of the turbidity and mixed liquids were used, the average relative error can also be shown in Table 2 (Er-NTU1-First peak; Er-NTU2-Second peak). According to the results of the relationship model between turbidity and absorbance (Figure 4), the average relative error is less than 40% (<100 NTU), less than 20% (100–200 NTU), and less than 5% (200–400 NTU), respectively (Table 2).

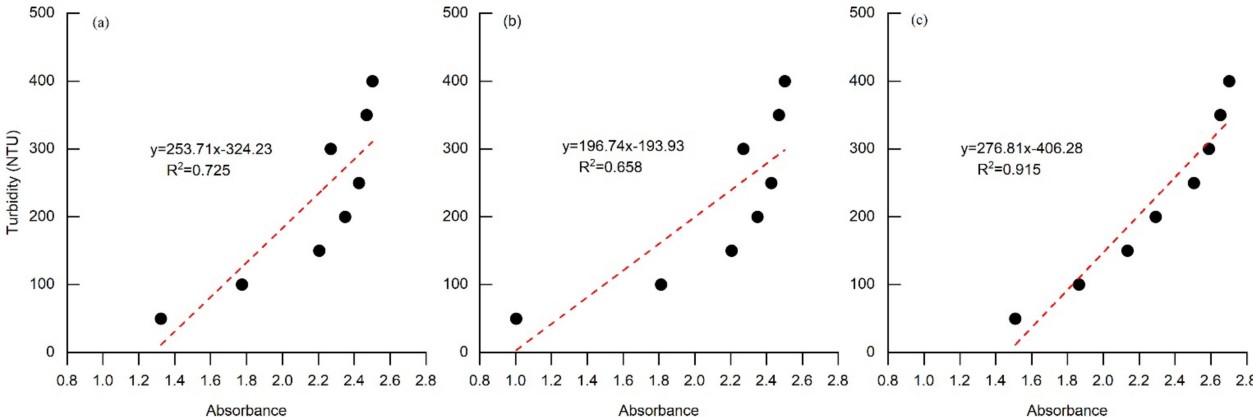

**Figure 3.** Calibration of turbidity using absorbance at the peak values. (**a**) First peak values of turbidity; (**b**) second peak values of turbidity; and (**c**) peak values of mixed liquids (turbidity and Chlorella).

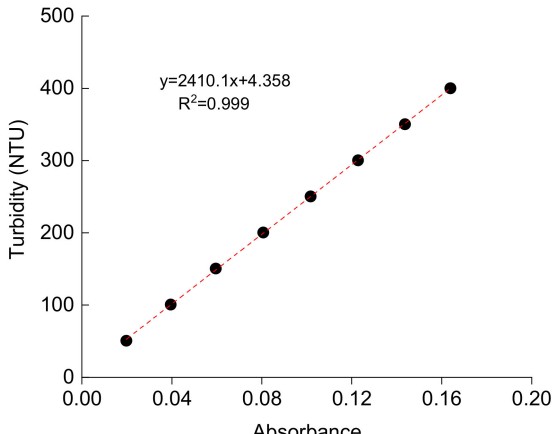

**Figure 4.** Calibration of turbidity using absorbance at 680 nm.

**Table 2.** Effect of Chlorella on turbidity measurement.

| NTU | C-A | M-A | Er-A | C-NTU1 | M-NTU1 | Er-NTU1 | C-NTU2 | M-NTU2 | Er-NTU2 |
| | 680 nm | 680 nm | 680 nm | First Peak | First Peak | First Peak | Second Peak | Second Peak | Second Peak |
|---|---|---|---|---|---|---|---|---|---|
| 50 | 0.058 | 0.117 | 50.18 | 3569 | 6329 | 40.61 | 2141 | 3433 | 37.65 |
| 100 | 0.064 | 0.137 | 53.41 | 4459 | 7424 | 39.94 | 3665 | 5381 | 31.88 |
| 150 | 0.075 | 0.157 | 51.85 | 4967 | 6329 | 21.51 | 4967 | 6329 | 21.51 |
| 200 | 0.090 | 0.178 | 49.24 | 5317 | 6397 | 16.88 | 5317 | 6397 | 16.88 |
| 250 | 0.106 | 0.199 | 46.91 | 5924 | 6524 | 9.20 | 5924 | 6524 | 9.20 |
| 300 | 0.119 | 0.220 | 45.69 | 6131 | 6515 | 5.88 | 6131 | 6515 | 5.88 |
| 350 | 0.132 | 0.241 | 45.33 | 6231 | 6438 | 3.21 | 6231 | 6438 | 3.21 |

Note: NTU: Mixed liquids; C-A, M-A, and Er-A are the measured value, model-predicted value, and relative error of the 680 nm spectral absorbance of the mixed liquid, respectively; C-NTU1 (measured value), M-NTU1 (model-predicted peak value), and Er-NTU1 (relative error) are the turbidity changes calibrated by the first absorbance peak in the mixture; C-NTU2 (measured value), M-NTU2 (model-predicted peak value), and Er-NTU2 (relative error) are the turbidity changes calibrated by the second absorbance peak in the mixture.

### 3.3. Turbidity Measurement

The measured absorbance of the experimental mixtures may be affected by various factors, such as the chromaticity of the Chlorella. Characteristic spectra sensitive to turbidity (680 nm) and Chlorella (240 nm) were selected, and then the experiments assessed the correlation between turbidity and the corresponding absorbance (Figure 5). This allowed

a turbidity measurement model to be established, which was then applied to predict the properties of mixed liquids. Figure 5 shows the relationship models NF240 and NF680 that relate turbidity and absorbance at 240 and 680 nm, respectively. The relationships CF240 and CF680 model the turbidity of the mixed liquids at the same respective spectral bands. Using these models, the theoretical and practical turbidities of the mixed solutions were analyzed (Table 3). The results show that (1) when the turbidity is below 200 NTU, Chlorella greatly influences the measured turbidity: The relative errors at 50 and 100 NTU are 73.07% and 57.06%, respectively (mean 65.07%). (2) When the turbidity is 250 NTU, the relative error is 8.34%, and at 400 NTU, it is only 2.18%. Overall, when the turbidity is low (<100 NTU), the relative error is large, and when the turbidity is high (greater than 250 NTU), the average relative error is below 5%.

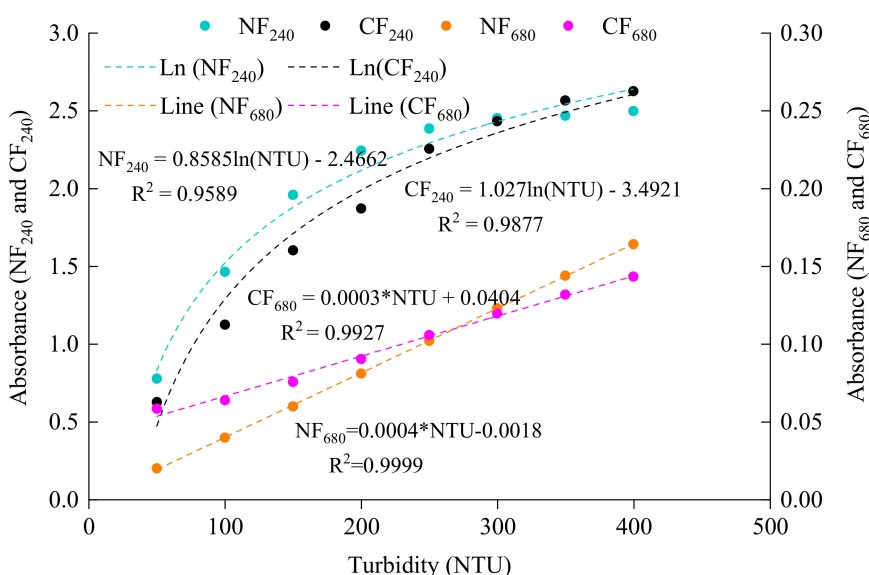

**Figure 5.** Calibration model of characteristic spectral absorbance and turbidity measurement. NF240 and NF680 are the model relationships between the turbidity and absorbance of turbidity standard solution at 240 and 680 nm, respectively. CF240 and CF680 are the model relationships between the turbidity of mixed liquids and absorbance at 240 and 680 nm, respectively.

**Table 3.** Turbidity correction and measurement error analysis.

| NTU | NF240 | NF680 | CF240 | CF680 | MF680 | NCE (%) |
|-----|-------|-------|-------|-------|-------|---------|
| 50  | 0.775 | 0.02  | 0.625 | 0.058 | 0.016 | 73.07   |
| 100 | 1.463 | 0.04  | 1.123 | 0.064 | 0.027 | 57.06   |
| 150 | 1.958 | 0.06  | 1.601 | 0.075 | 0.047 | 38.18   |
| 200 | 2.243 | 0.081 | 1.870 | 0.09  | 0.063 | 30.18   |
| 250 | 2.384 | 0.102 | 2.255 | 0.106 | 0.097 | 8.34    |
| 300 | 2.453 | 0.123 | 2.432 | 0.119 | 0.118 | 1.22    |
| 350 | 2.468 | 0.144 | 2.566 | 0.132 | 0.137 | 4.05    |
| 400 | 2.498 | 0.164 | 2.625 | 0.143 | 0.146 | 2.18    |

Note: NTU: Standard turbidity liquids; NF240: Absorbance of turbidity standard solution at 240 nm; NF680: Absorbance of turbidity standard solution at the spectral frequency of 680 nm; CF240: Measured absorbance of mixture at 240 nm; CF680: Measured absorbance of mixture at 680 nm; MF680: Predicted absorbance of mixture at 680 nm; NCE (%): Relative error between corrected value according to the model and measured value of the mixed liquid.

## 4. Discussion

### 4.1. Absorbance and Turbidity

The theoretical model in this paper is based on the Beer–Lambert law and the principle of additivity. It is assumed that the different components do not interfere with each other's absorption of light at a given frequency. Solutions of Chlorella, standard turbidity liquids, and mixtures of both were prepared, and the influence of Chlorella on turbidity measurements was analyzed by considering the maximum absorbance peaks and spectral changes among the three different types of liquid. Previous studies have shown that changing the concentration and turbidity will affect the measurement of other components, such as algae and chlorophyll [25]. The experiments reported here showed that Chlorella also influences the turbidity measurement (Tables 2 and 3). The relative error of theoretical and experimental values was high (>50%) when the turbidity was no more than 100 NTU and below 10% when the turbidity was more than 100 NTU (< 400 NTU). Figure 2b shows that after mixing Chlorella with the standard turbidity liquid, the double absorption peak became a single peak. In previous studies, in the low turbidity (<100 NTU), the absorbance value of turbidity standard solution will fluctuate to a certain extent and have different peaks [34,35]. In this experiment, when the turbidity is under 100 NTU, the absorbance of the turbidity solution also has a certain peak fluctuation (double peaks). When Chlorella is added to the turbidity solution, the bimodal phenomenon of its mixture disappears. The main reason is that the standard turbidity sample is made of organic substances. Most of the organics are absorbed in the ultraviolet region [35], and the peak absorbance of Chlorella is also in this region. This may be the reason why the bimodal phenomenon of mixed liquor disappears after the addition of Chlorella. Therefore, the addition of Chlorella affects the turbidity, especially the change in low turbidity absorbance, and the degree and factors of influence need further experiments and analysis.

The Beer–Lambert law assumes that the incident light is parallel and monochromatic and that the concentration of material in the measured water body is not high. A high concentration of material could facilitate the interaction between light-absorbing particles, thereby affecting the degree of absorption. Previous work [34] has established that when the absorbance peak is large (greater than 1), there may be interactions between particles of different components. However, the boundary between high and low turbidity is not clearly defined, although the boundary in the present work appears to be at around 100 NTU (Figure 2). The influence of particle size on the measurement results is not considered in the absorbance analysis. Subsequent experimental design could dilute high-turbidity liquids by a certain proportion to control the absorbance peak value below 1. This could be conducted by considering the particle size, changes in the suspended components, and the influence of different concentrations of Chlorella, etc.

### 4.2. Influence of Chlorella on Turbidity Measurement

The present results show that Chlorella greatly affected turbidity. The theoretical and practical values were different when measuring the absorbance of a standard turbidity solution and a mixture containing Chlorella. Many factors, including the absorption of colored soluble substances, particle size, temperature, and salinity of dissolved substances may affect the results [36,37]. When a spectrophotometer is used for analysis, the absorption curve is an important basis for selecting the wavelength of incident light in quantitative analysis. The most suitable wavelength is selected in the absorption spectrum of the sample, with higher accuracy and sensitivity; namely, by selecting the appropriate wavelength, the influence of the noise in the water sample on the measurement of turbidity can be eliminated.

If the surface of fine particles absorbed yellow organic matter, the measured absorbance will be greater than predicted. The peak absorption values for water containing a single factor or a mixture of components vary with the concentration. The yellow and green pigments contained in water have different degrees of influence on turbidity measurement. In this paper, the turbidity solution is calibrated as formazine, and a standard model is

established at 680 nm to eliminate the influence of different pigments in water [38]. In addition, the correlation between water absorbance and Chlorella content is stable after 680 nm. Compared with the other bands, it can better reflect the correlation between Chlorella and absorbance. Moreover, the phycobilin of Chlorella may have a certain impact on the absorption spectrum. Previous research [25] has shown that the absorbance of Chlorella has a stable and significant correlation with the content of chlorophyll a: The maximum correlation coefficient (0.999) was obtained at 240 nm. The present work considered the characteristic spectrum at 240 nm as the absorbance of Chlorella. When the turbidity standard curve was compiled by spectrophotometry, the absorbance at 680 nm was selected as the characteristic spectrum for turbidity measurement. From the relationship between the characteristic spectrum and the absorbance of water, the relationship between the characteristic spectra was modeled. This allowed the turbidity and changes in the water to be calculated and the possible influence of colored substances in Chlorella on the measurement results to be analyzed (Figure 5 and Table 3). Large errors were found at low turbidity: 73.07% and 57.06% at 50 and 100 NTU, respectively (Table 3). These errors were greater than the values of 37.65% and 39.46% (Table 2) that arose using the characteristic spectral turbidity calibration and absorbance peak measurement.

At present, information regarding water turbidity is obtained primarily through measuring absorbance, transmission or spectral scattering of a component-sensitive spectrum [36]. Any change in a certain sensitive spectral absorbance can reflect the change in turbidity of the water body. Due to the complex composition of environmental water, changes in turbidity could be caused by inorganic substances (such as silt or clay) or organic substances (such as algae, plankton, and decaying matter). In addition to these suspended solids, the turbidity can be caused by colored dissolved organic matter, fluorescent dissolved organic matter, and other dyes [36]. Furthermore, it may be affected by changes in conditions, such as temperature or salinity [37]. The present work used the absorbance peak and the corresponding characteristic spectrum to analyze changes in components in water and their concentrations; the influence was more significant at low turbidity. Therefore, calibration of the turbidity characteristic spectrum and absorbance peak calculation improved the results.

To improve the accuracy of turbidity measurement, it is necessary to combine the absorbance peak measurements for different components with comprehensive high-resolution spectral analysis techniques (i.e., spectral resolution of 0.17 nm or higher). The practical significance of this study lies in the applicability in the field of the relationship between the absorbance peak value and the turbidity of a certain substance or different substances that can be obtained through controlled experiments. Therefore, this study provides a basis for further study of algae, turbidity, and their interactive effects. At the same time, through calibration, the changes in turbidity of a specific spectrum can be measured on-site to calculate the concentrations of the changes in a certain component or substance. As a result, information regarding the transport or flux of different substances in the water body can be obtained.

## 5. Conclusions

This paper quantitatively analyzes the effect of mixing Chlorella with different standard turbidity liquids on the measurement of turbidity. The main conclusions are as follows: Chlorella greatly affects the measurement of turbidity. The relative error between the theoretical and experimental values is higher when the turbidity is lower ($\leq$100 NTU). As the turbidity increases above 200 NTU, the measurement error is less than 10%. Better results can be obtained by analyzing changes in turbidity and the effect of Chlorella on turbidity measurements using a combination of absorbance peaks and characteristic spectral calibration. At the same time, using the turbidity calibration model in this paper, the concentration and variation information of the characteristic suspended components in the water sample can be obtained. The results of this study help in improving the measurement

accuracy of field observation of different component concentrations and in facilitating the research and development of new turbidity sensors.

**Author Contributions:** W.Z. wrote the first draft of the manuscript and prepared the figures. D.Z. and Z.Z. conducted the lab experiment. J.S. data analysed and calculation. Y.W., X.W., Y.L. (Yang Lv), and Y.L. (Yue Li) improved the manuscript. B.S. and Y.L. (Youcai Liu) contributed to the design of the structure and improved the manuscript. All authors have read and agreed to the published version of the manuscript.

**Funding:** This study was supported by the Open Research Fund of Key Laboratory of Coastal Science and Integrated Management, Ministry of Natural Resources (2021COSIMZ001), the Resource Management Office of Hebei Provincial Department of Science and Technology (226Z3301G), and the Research Funds of Happiness Flower ECNU (20212110).

**Institutional Review Board Statement:** Not applicable.

**Informed Consent Statement:** Not applicable.

**Data Availability Statement:** Not applicable.

**Acknowledgments:** The authors thank Cui H., Li J.J., Liu X.Y., Zhou H., and Zhang X.M. for providing the technical support for the experiment.

**Conflicts of Interest:** The authors declare no conflict of interest.

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
