# Peer review of "Laboratory Experiments to Assess the Effect of Chlorella on Turbidity Estimation"

_water, doi:10.3390/w14193184_

Round 1
Reviewer 1 Report
1. Lines 79-82: The purposes of this study are described in MS, however many unsolved questions still exist in discussion. For example, how the factors affect the turbidity alternation, facilitating us for improving the turbidity measurement and providing technical support through calibration. Authors have to re-organize the motivation and purposes of this study to meet the observation of experiments.
2. What are the innovation and strengths of this study? Complex chemical-physical interactions cause the uncertainty of turbidity accuracy, why Chlorella was collected as a research target?
3. Figure 1 is difficult to understand what the authors try to present.
4. Line 96: Averaged values are used for further discussion in this study. However, QA/QC have to be clearly described to ensure the representative of experiment outcomes.
5. Line 120: Should be "log".
6. Table 1: What is the unit of sample 9? If that is turbidity, then the first column need to be modified.
7. Figures 2 and 3: The peak heights are used for establishing correlation between absorbance and turbidity. Is any possible try to use the areas below the curves? It might have better correlations.
8. Lines 160-161, 184: A specific 680nm spectrum was adopted for calibration. What spectrum is used in experiments of section 3.1? How to decide this optimal spectrum?
9. First paragragh of section 4.1 should be motivation of this study.
10. Lines 253-260, 273-274: There are too many possibilities and uncertainties to explain the results of experiments. Related references have to be cited for discussion, rather leave these issues as future works.
11. Line 268: How to prove the uniform particle size?
12. Line 319: of and?
Author Response
Comments and Suggestions for Authors
- Lines 79-82: The purposes of this study are described in MS, however many unsolved questions still exist in discussion. For example, how the factors affect the turbidity alternation, facilitating us for improving the turbidity measurement and providing technical support through calibration. Authors have to re-organize the motivation and purposes of this study to meet the observation of experiments.
Reply to reviewer:
According to the comments and requirements of the reviewers, the original Line 79-82 is revised again. See the revised text Line 76-86.
- What are the innovation and strengths of this study? Complex chemical-physical interactions cause the uncertainty of turbidity accuracy, why Chlorella was collected as a research target?
Reply to reviewer:
As the reviewer commones, the complex chemical and physical processes in the field lead to great uncertainty in turbidity measurement, and different influence factors are difficult to control. The motivation of this paper is to define the influence of other factors in water on turbidity measurement through Lab control experiments and improve the accuracy of turbidity measurement. At the same time, it provides the technical support for improving the precision of optical measurement of suspended solids concentration in water colume.
Take Chlorella as an example, this paper analyzes the change of absorbance. Through Lab experiments, the effect of Chlorella on different turbidity liquids was done. Chlorella is a phylum of green algae, which is widely distributed. It is one of the earliest life on the earth and appeared more than 2 billion years ago. It is a highly efficient photosynthetic plant. It grows and reproduces by photosynthetic autotrophy. In the south area of China, some surface water bodies often have a large number of algae (including Chlorella) in summer and autumn.
- Figure 1 is difficult to understand what the authors try to present.
Reply to reviewer:
According to the reminder of the reviewer, figure 1 has been revised. The author's intention is to express the physical characteristics of Chlorella used in this experiment, including particle size and shape, because the composition, shape and size of suspended solids in water can affect optical sensors or optical measurements (although this article does not discuss this aspect), and provide a reference for later experiments and research.
1.4 Line 96: Averaged values are used for further discussion in this study. However, QA/QC have to be clearly described to ensure the representative of experiment outcomes.
Reply to reviewer:
The spectrophotometer used in the experiment was tested and calibrated by the manufacturer (Line 98-99).
During the experiment, each sample was uniformly mixed, the number and time was unified. In order to reduce the human error in the measurement process, three groups of stable data were selected from five groups of data for average and experimental analysis (Line 116-119).
- Line 120: Should be "log".
Reply to reviewer:
Line 134,Line 141 in the revised text: "lg" is right.
- Table 1: What is the unit of sample 9? If that is turbidity, then the first column need to be modified.
Reply to reviewer:
Table 1 was revised according to the suggestions of the reviewers (Line 103-106; Line 126)
- Figures 2 and 3: The peak heights are used for establishing correlation between absorbance and turbidity. Is any possible try to use the areas below the curves? It might have better correlations.
Reply to reviewer:
Thank the reviewer for the suggestions. Spectrophotometer measure the absorbance peak of specific spectrum of suspended solids in water, and obtains the change information of different components in the water samples. The correlation may be better through the curve area analysis. In this paper, the Spectrophotometer was used to measure sbsorbance peak values and the change information of the samples. Correlation analysis by turbidity versus absorbance peaks can better reflect the real state of the experiment, and its correlation change is more sensitive.
- Lines 160-161, 184: A specific 680nm spectrum was adopted for calibration. What spectrum is used in experiments of section 3.1? How to decide this optimal spectrum?
Reply to reviewer:
Line 291-296
When spectrophotometer is used for analysis, the absorption curve is an important basis for selecting the wavelength of incident light in quantitative analysis. The most suitable wavelength is selected in the absorption spectrum of the sample, with high accuracy. That is, by selecting the appropriate wavelength, the influence of the noise in the water sample on the measurement of turbidity can be eliminated.
Line 299-305
The yellow and green pigments contained in water have different degrees of influence on turbidity measurement. In this paper, the turbidity solution is calibrated as formazine, and a standard model is established at 680nm to eliminate the influence of different pigments in water. In addition, the correlation between water absorbance and Chlorella content is stable after 680nm. Compared with other bands, it can better reflect the correlation between Chlorella and absorbance.
1.9 First paragragh of section 4.1 should be motivation of this study.
Reply to reviewer:
Thanks for the suggestions. The original text has been revised. The content of 4.1 is adjusted to the first part (Line 76-86)
1.10 Lines 253-260, 273-274: There are too many possibilities and uncertainties to explain the results of experiments. Related references have to be cited for discussion, rather leave these issues as future works.
Reply to reviewer:
Thanks for the suggestions. The original text has been revised (Line 263-273).
In previous studies, in the low turbidity (< 100 NTU), the absorbance value of turbidity standard solution will fluctuate to a certain extent and have different peaks [34,35]. In this experiment, when the turbidity under 100 NTU, the absorbance of the turbidity solution also has a certain peak fluctuation (double peaks); When Chlorella is added to the turbidity solution, the bimodal phenomenon of its mixture disappears. The main reason is the standard turbidity sample is made of organic substances. Most of the or-ganics are absorbed in the ultraviolet region [35], and the peak absorbance of Chlorella is also in this region. This may be the reason why the bimodal phenomenon of mixed liquor disappears after the addition of Chlorella. Thus, the addition of Chlorella affects the turbidity, especially the change of low turbidity absorbance and the degree and factors of influence need further experiments and analysis.
1.11 Line 268: How to prove the uniform particle size?
Reply to reviewer:
Thanks for the suggestions. Fig. 1 can be seen the Chlorella is round in shape, and the particle size is relatively uniform. the author wants to express that (1) the particle size of Chlorella is larger than 2um, which is a component of suspended solids and affects the turbidity measurement results; (2) Chlorella is round, because the composition, morphology and size of suspended particles will affect the measurement results. This aspect is not discussed in this paper, but it may have some significance for subsequent related experiments.
1.12 Line 319: of and?
Reply to reviewer:
It has been revised according to the suggestions of the reviewers (Line 341).

Reviewer 2 Report
Presented manuscript entitled “Laboratory experiments to assess the effect of Chlorella on turbidity estimation” is written with a very good English and, in my opinion, no additional corrections are not needed in this field.
Below, please find my detailed comments and questions.
· In title word “Chlorella” is written in italics, however later in the text not. Please unify.
· Keywords “Spectrophotometer” and “Absorbance” are, in my opinion, too general and do not exactly fit. Maybe “water quality” or “modeling” will be better?
· Please change the citations from [10,11,12,13], [14,15,16,17,18,19] and [20,21,22,23,24] to [10-13], [14-19] and [20-24].
· Please add information about the company of used spectrophotometer and optical microscope. What was the magnification used for microscopic imaging?
· Please add more information about preparing the turbidity standard solutions via ISO 7027 method.
· The scale in Figure 1a is almost invisible. Please consider its enlargement.
· In Table 1 is the value 2.55 the turbidity of chlorella solution? What is the meaning of value 107 N/L?
· Figure 2: Please consider changing the colors of absorbance spectra in such a way, that same turbidity value on both spectra has the same color. It will make it easier to understand.
· Please explain more deeply, why the 680 nm wavelength was taken for analysis?
· Why there is only calibration curve for 680 nm, not for first and second absorbance peaks also?
· Authors should consider adding to Table 2 and 3 additional headers, dividing the Tables’ content into smaller groups (e.g., 680 nm, first/second absorbance peak) for better recognition.
· For sections Discussion and Conclusions, I do not have any comments or questions. They are written very well and in understandable way.
Concluding, I suggest a major revision of presented manuscript with perspective of acceptance after some corrections.
Author Response
Comments and Suggestions for Authors
- In title word “Chlorella” is written in italics, however later in the text not. Please unify.
Reply to reviewer:
Thank the reviewer for suggestion, the original text has been revised (Line 2).
- Keywords “Spectrophotometer” and “Absorbance” are, in my opinion, too general and do not exactly fit. Maybe “water quality” or “modeling” will be better?
Reply to reviewer:
According to the suggestions of the reviewer, the original text has been revised (Line 38)
2.3 Please change the citations from [10,11,12,13], [14,15,16,17,18,19] and [20,21,22,23,24] to [10-13], [14-19] and [20-24].
Reply to reviewer:
According to the suggestions of the reviewer, the original text has been revised (Line 62-65)
2.4 Please add information about the company of used spectrophotometer and optical microscope. What was the magnification used for microscopic imaging?
Reply to reviewer:
According to the suggestions of the reviewer, the original text has been revised (Line 97-98; Line 101-103)
2.5. Please add more information about preparing the turbidity standard solutions via ISO 7027 method.
Reply to reviewer:
Thank the reviewer for suggestion, the original text has been revised (Line 107-113)
2.6 The scale in Figure 1a is almost invisible. Please consider its enlargement.
Reply to reviewer:
According to the suggestions of the reviewer, the original text has been revised (Fig.1). (Line 123-124)
2.7 In Table 1 is the value 2.55 the turbidity of chlorella solution? What is the meaning of value 107 N/L?
Reply to reviewer:
According to the suggestions of the reviewer, the original text has been revised (Line103-105)
2.8 Figure 2: Please consider changing the colors of absorbance spectra in such a way, that same turbidity value on both spectra has the same color. It will make it easier to understand.
Reply to reviewer:
According to the suggestions of the reviewer, the original text has been revised (Line 161-162)
2.9 Please explain more deeply, why the 680 nm wavelength was taken for analysis?
Reply to reviewer:
According to the suggestions of the reviewer, The relevant explanations are as follows:
Line 291-296
When spectrophotometer is used for analysis, the absorption curve is an important basis for selecting the wavelength of incident light in quantitative analysis. The most suitable wavelength is selected in the absorption spectrum of the sample, with high accuracy. That is, by selecting the appropriate wavelength, the influence of the noise in the water sample on the measurement of turbidity can be eliminated.
Line 299-305
The yellow and green pigments contained in water have different degrees of influence on turbidity measurement. In this paper, the turbidity solution is calibrated as formazine, and a standard model is established at 680nm to eliminate the influence of different pigments in water. In addition, the correlation between water absorbance and Chlorella content is stable after 680nm. Compared with other bands, it can better reflect the correlation between Chlorella and absorbance.
2.10 Why there is only calibration curve for 680 nm, not for first and second absorbance peaks also?
The purpose of this paper is to study the impact of Chlorella on the turbidity measurement results. The purpose of using 680nm is to calibrate the first peak and the second peak, without considering the possible impact of the chromaticity of water on the turbidity measurement results. The reason why 680nm is selected has been explained in 1.8. In this paper, the first and second peak absorbance and turbidity of different turbidity solutions have been calibrated, and the calibration results of different turbidity solutions (Fig. 3 a, b) and Chlorella and mixed solutions (Fig. 3c) have been analyzed. The calculation results are shown in Table 2. According to the comments of the reviewers, the structure of the original text has been adjusted to a certain extent (Line183-184).
2.11 Authors should consider adding to Table 2 and 3 additional headers, dividing the Tables’ content into smaller groups (e.g., 680 nm, first/second absorbance peak) for better recognition.
Reply to reviewer:
According to the suggestions of the reviewer, the original text has been revised (Table 2) (Line194-195)
2.12 For sections Discussion and Conclusions, I do not have any comments or questions. They are written very well and in understandable way.
Thank the reviewer for comments.
13.Concluding, I suggest a major revision of presented manuscript with perspective of acceptance after some corrections.
Reply to reviewer:
According to the suggestions of the reviewer, the drawings, tables, contents and structures in the original text have been revised according to the requirements of the reviewers.
Round 2
Reviewer 1 Report
Authors have revised the manuscript based on my suggestions. I have mo more suggestions.
Author Response
Thank the reviewer for the suggestions and hard work!
Reviewer 2 Report
Authors corrected the manuscript (although the version they applied is hardly readable due to to change tracking mode).
In this regard, I suggest acceptance of presented manuscript in the revised form.
Author Response

(The authors gave the same response as above.)
